# Evaluation of Physiological Parameters of Intestinal Sulfate-Reducing Bacteria Isolated from Patients Suffering from IBD and Healthy People

**DOI:** 10.3390/jcm9061920

**Published:** 2020-06-19

**Authors:** Ivan Kushkevych, Jorge Castro Sangrador, Dani Dordević, Monika Rozehnalová, Martin Černý, Roman Fafula, Monika Vítězová, Simon K.-M. R. Rittmann

**Affiliations:** 1Department of Experimental Biology, Faculty of Science, Masaryk University, Kamenice 753/5, 62500 Brno, Czech Republic; jorgecas@usal.es (J.C.S.); cernyarchaea@mail.muni.cz (M.C.); vitezova@sci.muni.cz (M.V.); 2Faculty of Biology, University of Salamanca, Campus Miguel de Unamuno C/Donantes de sangre, s/n 37007 Salamanca, Spain; 3Department of Plant Origin Foodstuffs Hygiene and Technology, Faculty of Veterinary Hygiene and Ecology, University of Veterinary and Pharmaceutical Sciences, 61242 Brno, Czech Republic; dani_dordevic@yahoo.com; 4Centre of Region Hana for Biotechnological an Agricultural Research, Central Laboratories and Research Support, Faculty of Science, Palacky University Olomouc, 77900 Olomouc, Czech Republic; Jarosova.Monika@email.cz; 5Department of Medical Biophysics, Danylo Halytsky Lviv National Medical University, 69 Pekarska St., 79010 Lviv, Ukraine; roman_fafula@ukr.net; 6Archaea Physiology & Biotechnology Group, Department of Functional and Evolutionary Ecology, Universität Wien, Althanstraße 14, 1090 Vienna, Austria

**Keywords:** ulcerative colitis, intestinal microbiota, bowel disease, sulfate reduction, hydrogen sulfide, toxicity

## Abstract

Background: Inflammatory bowel diseases (IBDs) are multifactorial illnesses of the intestine, to which microorganisms are contributing. Among the contributing microorganisms, sulfate-reducing bacteria (SRB) are suggested to be involved in the process of bowel inflammation due to the production of hydrogen sulfide (H_2_S) by dissimilatory sulfate reduction. The aims of our research were to physiologically examine SRB in fecal samples of patients with IBD and a control group, their identification, the study of the process of dissimilatory sulfate reduction (sulfate consumption and H_2_S production) and biomass accumulation. Determination of biogenic elements of the SRB and evaluation of obtained parameters by using statistical methods were also included in the research. The material for the research consisted of 14 fecal samples, which was obtained from patients and control subjects. Methods: Microscopic techniques, microbiological, biochemical, biophysical methods and statistical analysis were included. Results: Colonies of SRB were isolated from all the fecal samples, and subsequently, 35 strains were obtained. Vibrio-shaped cells stained Gram-negative were dominant in all purified studied strains. All strains had a high percentage of similarity by the 16S rRNA gene with deposited sequences in GenBank of *Desulfovibrio vulgaris*. Cluster analysis of sulfate reduction parameters allowed the grouping of SRB strains. Significant (*p* < 0.05) differences were not observed between healthy individuals and patients with IBD with regard to sulfate reduction parameters (sulfate consumption, H_2_S and biomass accumulation). Moreover, we found that manganese and iron contents in the cell extracts are higher among healthy individuals in comparison to unhealthy individuals that have an intestinal bowel disease, especially ulcerative colitis. Conclusions: The observations obtained from studying SRB emphasize differences in the intestinal microbial processes of healthy and unhealthy people.

## 1. Introduction

Inflammatory bowel diseases (IBD) are illnesses whose incidence have increased in recent years in European countries [1,2,3,4]. IBD include two types of conditions: Crohn’s disease (CD) and ulcerative colitis (UC) [2,3]. The etiology of these diseases remains uncertain, and it is not known how environmental, genetic, microbial, cellular and molecular factors involved cooperate to make IBD happen [5]. A microbiological origin has been suggested, as the illnesses respond to the treatment with antibiotics, but no conclusive evidence has been found [6,7,8,9,10]. Symptomatology of IBD, in general, comprises watery diarrhea with blood, weight loss, abdominal pain, increased gut permeability, an urgency to defecate, arthritic effects and general discomfort [7,11].

Attempts to address a microbial origin to IBD by traditional techniques have failed. The more powerful explanation for the disease in humans is believed to be the genetic predisposition for impairment of the immune response, acting against antigens of the normal microbiota [3,5]. However, research has provided much evidence on the relationship between microorganisms and IBD and the involvement of bacterial sulfate metabolism on them [12,13,14,15,16]. Studies of the intestinal microbiota of UC patients and especially the study with animal models of gut inflammation demonstrate the relevance of sulfate-reducing bacteria (SRB) for the beginning and the prevalence of the process of inflammation [11,17,18].

An increased level of SRB and the final products of their metabolism (hydrogen sulfide (H_2_S) and acetate) are always detectable in the feces of patients with IBD [19,20,21,22,23,24,25,26,27]. SRB use sulfate as an electron acceptor in the process of dissimilatory sulfate reduction (DSR). The final product of this process is H_2_S, which can be toxic, mutagenic and cancerogenic [19]. For this process, electron donors such as molecular hydrogen (H_2_) [28,29] or a varied range of organic compounds are necessary. Organic compounds (e.g., lactate, pyruvate, ethanol, amino acids, etc.) can be used by SRB as a carbon and energy source [30,31,32]. These compounds are usually oxidized to acetate, which can be an additional factor involved in IBD development [7,9,33].

The dominant species of SRB in the human intestine belong to *Desulfovibrio* sp. [7,12,34,35]. The studies emphasize connections between the presence of SRB in the intestines and the prevalence of ailments, such as cholecystitis, brain abscesses and abdominal cavity ulcerative enterocolitis [7,11,36,37]. SRB are not the only organisms of the intestine producing H_2_S. Although SRB isolates coming from the intestine of humans have been reported to belong almost exclusively to *Desulfovibrio* sp., other genera are also part of the SRB microbiota: *Desulfotomaculum, Desulfobulbus, Desulfomicrobium, Desulfomonas, Desulfonema* [12,28]. In the intestine, SRB, mainly *Desulfovibrio* sp., form biofilms together with other intestinal bacteria, such as species of *Bacteroides, Pseudomonas, Clostridium* and *Escherichia*. *Desulfovibrio* spp. are also present in a certain percentage of the healthy population [38,39,40]. They were first isolated from human feces in 1976 by W.E.C. Moore and identified as *Desulfomonas pigra* [41], which was then reclassified as *Desulfovibrio piger* [34]. *Desulfovibrio* belongs to *Desulfovibrionaceae* [42]. SRB are present in various environments and they make a high impact on animal and human health since their presence is a possible contributing factor in the development of inflammatory bowel diseases [20].

The aim of the research was to detect SRB in fecal samples of patients with IBD and healthy people, to purify these cultures and to identify them (phylogenetically by 16S rRNA gene), to compare their morphology and to construct a phylogenetic tree based on the obtained sequences as well as to study the process of dissimilatory sulfate reduction (sulfate consumption and H_2_S production) and biomass accumulation. Determination of important trace elements for their metabolism and evaluation of obtained parameters by using statistical methods will as well be sought in this research.

## 2. Experimental Section

The fecal samples from healthy people (4 samples) and patients with IBD (10 samples) were collected. A summary of the features of the patients and healthy subjects used in this study is presented in Table 1 and Table 2. All the bacterial strains, which were isolated in the frame of this study, were purified. Fecal samples of 10 patients with IBD were treated in the University Hospital of Ostrava (Czech Republic) and 4 fecal samples of healthy subjects were collected in the same institution. The latter was used in the present study as the controls. All subjects gave their informed consent for inclusion before they participated in the study. The study was conducted in accordance with the Declaration of Helsinki, and the protocol was approved by the Ethics Committee of University Hospital Ostrava and Masaryk University (551/2018). All bacterial strains were kept in the collection of microorganisms in the Laboratory of Anaerobic Microorganisms of the Department of Experimental Biology at Masaryk University (Brno, Czech Republic).

All donors who were suffering from IBD at the given time were undergoing therapy (with exception of a donor of sample M-02). Prescribed therapeutics which served for attenuation and management of the disease are summarized in Table 2.

### 2.1. Bacterial Culture Isolation, Purification and Cultivation

Intestinal SRB strains were grown in the SRB medium modified by Kováč and Kushkevych (2016) [43] based on the cultivation of Postgate medium for SRB [35]. Ascorbic acid solution (10% (*w*/*v*)) was prepared and added into the medium to a final concentration of 0.1 g L^−1^. Ascorbic acid is an important compound for decreasing the redox potential (E_h_ = −100 mV). For the detection of growing colonies, a 10% (*w*/*v*) of Mohr’s salt solution (NH_4_)_2_Fe(SO_4_) × 6H_2_O was added to a final concentration of 10 mL solution per L of the medium. Iron ions included in Mohr’s salt react with H_2_S produced by SRB cultures forming a black precipitate (FeS), which allows visualizing the SRB colonies. After completion, the pH of the medium was adjusted to approximately 7.5 with a 1 mol L^−1^ NaOH solution. For the cultivation in Petri plates, the same composition of medium was used, with the addition of nutrient agar in a concentration of 20 g L^−1^ before the sterilization. For obtaining pure SRB cultures, isolation and purification procedure was carried out. The present experiment was started from the SRB fecal samples from the described in Table 1 subjects. 

Positive samples containing SRB were diluted in a series of tube tests containing modified SRB liquid medium. The isolated colonies were suspended in an Eppendorf containing modified SRB liquid medium. The newly isolated and suspended colonies were cultivated at 37 °C, and after 5 d, precipitation of blacked ferrous sulfide indicated the proper growth of SRB. This whole process was repeated 3–5 times to assure a complete purification of SRB from other satellite microorganisms. The purification process was systematically monitored with the light microscope by the Gram staining of the isolated colonies after the sequential steps of isolation.

### 2.2. DNA Isolation, 16S rDNA Amplification and Sequence Analysis

Isolation of the bacterial DNA was a procedure that shall only be carried out when the isolated colonies have passed the purification process so that the extracted DNA was uniformed. A simple technique was used with the samples concerning this study: liquid cultures have a total volume of 1 mL. Firstly, for every purified sample that has undergone cultivation for three days, 200 µL were separated into Eppendorf tubes and centrifuged at 13,000× *g* to discard the supernatant. Then, biomass was suspended in 50 µL of sterile deionized water. 

The resuspended colonies were disrupted by heating at 98 °C for three min in an Eppendorf block heater. Afterwards, the samples with the disrupted cells were centrifuged at 13,000× *g* for 3 min. After this procedure, the supernatant remaining after the centrifugation step already contains the DNA of interest. The unequivocal identification of the cultivated strains was obtained by the 16S rDNA sequencing technique. For the amplification of the fragment of interest, 2 µL of the DNA extract of each sample was put together in a 300 µL Eppendorf with 1 µL of Taq PCR Master Mix Kit (Cat. No. 201445) and 0.1 µL each of a pair of universal primers: forward primer 8FPL 100 µmol L^−1^ (5′-AGT-TTG-ATC-CTG-GCT-CAG-3′) and reverse primer 806R 100 mol L^−1^ (5′-GGA-CTA-CCA-GGG-TAT-CTAAT-3′) [44,45]. 

PCR was carried out in a Schoeller PCR Thermocycler with a gradient (Labcycler Gradient), applying the program with 35 cycles. To check the outcome of the PCR, all reaction products were put through electrophoresis in 1.5% (*w*/*v*) agarose gel. Preparation of one of the said gels requires the mix of 75 mL or TAE (Tris-acetate-EDTA) or TBA (Tri-borate-EDTA) buffer with 1 g of agarose. The mixture is boiled up in a microwave until the whole solution acquires transparency. After boiling, and advisably a few minutes of cooling down later, Gelred^®^ Nucleic Acid Gel Stain was added in a proportion of 1 µL of dye per mL of the gel prepared. 3 µL of each PCR product was mixed with approximately and an equal volume of a 6× DNA loading dye and loaded into a well. Electrophoresis was run for 1 h at a tension of 80 V.

The purification of the PCR products was conducted via FastGene^®^ Gel/PCR Extraction Kit Cat. No. FG-91202. PCR products were afterwards sent to sequence analysis of the 16S rRNA gene fragment amplified. The obtained results of sequences were launched in BLAST, compared with already existing sequences of SRB and eventually uploaded in GenBank under the following accession numbers: MT027899, MT093800, MT093826, MT093820, MT093819, MT093823, MT093825, MT093830, MT093829, MT093831.

### 2.3. Measurement of Sulfate Consumption

The sulfate concentration in the liquid medium was measured by the turbidimetric method after time intervals of the cultivation. In total, 40 mg L^−1^ BaCl_2_ solution was prepared in 0.12 mol L^−1^ HCl and mixed with glycerol in a 1:1 ratio. The supernatant of the sample was obtained by centrifugation at 5000× *g* at 23 °C and 1 mL was added to 10 mL of BaCl_2_: glycerol solution and carefully mixed as described before [46]. The absorbance of the mixed solution was measured after 10 min at 520 nm (Spectrasonic Genesis 5, Ecublens, Switzerland). A cultivation medium without bacteria growth was used as a control.

### 2.4. Measurement of Hydrogen Sulfide Production

Measurement of the H_2_S produced in every sample was performed using the methylene blue method [47]. According to this procedure, 1 mL of cell-free culture medium of each centrifuged sample was pipetted into 10 mL of an aqueous solution of 5 g L^−1^ zinc acetate. Two milliliters of a solution of 0.75 g L^−1^ of p-amino dimethylaniline in 2 mol L^−1^ sulfuric solution was added, and the solutions were standing for 5 min at room temperature. Thereafter, 0.5 mL of 12 g L^−1^ ferric chloride solution in 15 mmol L^−1^ sulfuric acid solution was added, and the final mixture was standing for another 5 min and was centrifuged at 5000× *g* at 23 °C for 10 min. Parallel, the procedure was repeated with the initial addition of distilled water to have a control tube. After the procedure, the absorbance of the samples was measured at a wavelength of 665 nm [47].

### 2.5. Determination of Biomass Concentration

In total, 1 mL of liquid medium without Mohr’s salt in a plastic cuvette was measured in a biophotometer (Eppendorf BioPhotometer^®^D30, Hamburg, Germany) as a blank. The same procedure with the bacterial suspension was performed. The optical density (OD_340_) was always measured before the experiments to provide approximately the same amount of SRB [27].

### 2.6. Statistical Analysis

Statistical calculations of the results were carried out using the MS Office (2010), Origin 8.0 (https://www.originlab.com/) and Statistica 13 (http://www.statsoft.com/) software programs. Using the experimental data, the basic statistical parameters (mean: M, standard error: m, M ± SE) were calculated. The accurate approximation was when *p* ≤ 0.05 [48].

Hierarchical cluster analysis was applied to experimental data of measured variables by Ward’s method with Statistica software version 13.5.0.17, in order to classify experimental strains into groups of similar subjects. The cluster analyses were conducted by the inclusion of the following parameters: sulfate consumption, H_2_S production, biomass accumulation, and all parameters together as well as concentration data of physiological important elements (Na^+^, K^+^, Ca^2+^, Fe^2+^, Cu^2+^, Mg^2+^, Mn^2+^ separately and all described elements together). Cluster analysis gave overall differences among compared groups. Statistically significant differences between strains coming from healthy and unhealthy people were sought by one-way ANOVA for those variables that distribute normally, while a nonparametric test (Mann–Whitney U test) was performed for non-normally distributed variables. Both tests were conducted with IBM SPSS Statistics package, version 23. A *p*-value of less than 0.05 was considered significant.

## 3. Results

### 3.1. Microscopic Analysis of the Cultures

From the 14 fecal samples originally studied, 35 strains were isolated and purified. Out of these 35 strains, 24 were obtained from patients with IBD and the remaining 11 were purified from healthy subjects. Morphological analysis was performed on SRB isolates from both groups, showing that no differences were found in cellular shape. Vibrio-shaped cells stained Gram-negative were present in all purified strains (Figure 1).

### 3.2. Sequence Analysis

Sequence analysis of 16S rDNA allowed the unequivocal identification of the strains. All sequences of the studied strains were compared with SRB species uploaded in NCBI and identified as *Desulfovibrio vulgaris*. Accession number, amplicon lengths and similarities with reference strains of *Desulfovibrio* genus are presented in Table 3. Obtained sequences of the 16S rRNA gene from strains isolated from healthy and not healthy persons were compared with existing sequences of intestinal SRB from GenBank by Multiple Sequence Alignment (MSA) and a phylogenetic tree was constructed (Figure 2). Isolated and purified strains belong to one cluster in which they are grouped with *Desulfovibrio longreachensis* ACM 3958 and *Desulfovibrio vulgaris* subsp. Oxamicus DSM 1925. Other intestinal SRB species were grouped in more distant clusters.

### 3.3. Sulfate Consumption, H_2_S Production and Biomass Accumulation

Ten bacterial strains from healthy and unhealthy persons were compared by the parameters sulfate consumption, H_2_S production and biomass accumulation (Figure 3). The highest sulfate consumption was observed in all bacterial strains after 24–48 h. The highest rate of this consumption was detected for SRB Strain 1, Strain 2 and Strain 7 (83–90%, 84–89%, 85–91%, respectively). The lower level of sulfate consumption within the first 24 h was observed for Strain 4, Strain 5, Strain 9 and others. In this case, the slowest sulfate consumption (56%) was detected in Strain 4. It should be noted that the sulfate level was stable and almost constant in the medium of all strains after 48 h of cultivation, regardless of their provenience.

The other important parameter is the production of H_2_S converted from sulfate in the process of DSR. A peak of its production was detected at 24 h of cultivation for every strain. The highest production of H_2_S was observed for the following bacterial strains: Strain 4 (35 µmol g^−1^), Strain 5 (40 µmol g^−1^), Strain 6 (37 µmol g^−1^). Due to its toxicity at higher concentrations, production of H_2_S was halted when the highest peak was reached, while biomass continued to increase. The relation of H_2_S production per gram of biomass is consequently minor at 48 h of cultivation for each strain. The differences in biomass accumulation were not found to be statistically significant between strains coming from healthy and unhealthy individuals. The highest accumulation of biomass was observed for Strain 7 (fivefold increase at 48 h compared with initial concentration after 5 h of cultivation). The metabolisms’ balance during the initial 24 h of bacterial cultivation was examined. Some percentage of the consumed sulfate was utterly metabolized and was used in the production of H_2_S while the rest was temporarily stored in the form of the metabolic intermediates: APS and SO_3_^2−^ (Figure 4).

The left scale indicates the percentage of sulfate consumed with respect to the initial concentration of the inorganic compound in the cultivation media, showing that less than 50% of it was consumed within the first 24 h. Thus, the majority of it remained free in the medium.

### 3.4. Statistical and Cluster Analysis of Sulfate Reduction Parameters and Trace Elements

Sulfate reduction parameters and biomass accumulation (during 48 h) of intestinal SRB strains among healthy individuals and patients with IBDs are shown in Table 4. Statistically significant (*p* < 0.05) differences in the abovementioned parameters were not found between healthy and unhealthy individuals. All measured parameters (sulfate, H_2_S and biomass) after 5 h had statistically significant (*p* < 0.05) different values in comparison with values measured after 24 and 48 h, both among healthy individuals and patients with IBD.

Multivariate analysis of the results obtained from sulfate consumption, H_2_S production and biomass accumulation were performed. Cluster analysis is conducted regarding each of these variables separately and combined (Figure 5); with the aim of sorting the different isolated strains into groups according to the measured variables.

Clusters built from sulfate consumption and biomass accumulation data show that strains were clustered in the three main groups in both cases, apparently regardless of their origin (unhealthy or control subject). Rapid biomass accumulation has been observed for every strain, as was illustrated in Figure 3, notwithstanding, strains that yielded a less accumulation of biomass (*D. vulgaris* M9.2, *D. vulgaris* Z9.3, and *D. vulgaris* Z11.1) form one of the three main groups mentioned. Nevertheless, clusters obtained from H_2_S production data drop interesting results. Four of the strains obtained from subjects with IBD cluster together in two different clusters, while the remaining strain from an unhealthy person, *D. vulgaris* M9.3 cluster together with the *D. vulgaris* Z8.2 strain, both strains represented the highest producers, closely followed by *D. vulgaris* M8 and *D. vulgaris* M9.2. Strain *D. vulgaris* Z8.2 was an H_2_S overproducer strain coming from a healthy subject, who might be potentially more susceptible to IBD than the other control subjects due to the presented results.

Data obtained regarding intracellular trace element concentration present in the 35 strains isolated were used for cluster analysis. The consistency of the results relies on the fact that measurements were taken with six repetitions per strain. 

Analysis based on iron concentration shown in Figure 6 drops the formation of two clusters, first of each is mostly conformed by strains isolated from patients with IBD. Precisely, the first subcluster of this group includes 15 of these strains and just one strain was originating from a healthy subject. Results for Na^+^ and K^+^ showed two clusters, and the ones for calcium formed three clusters. All of them show a more equal proportion of both types of strains, bearing in mind that there is a considerably higher number of strains isolated from unhealthy patients.

For copper, the first subcluster was observed for those strains that do not have a detectable concentration of the element (five from patients and two from healthy subjects). Remaining strains equally distributed in the other subclusters, as it happens with Mg^2+^, finding two main clusters for both elements (Figure 7). Clusters based on Mn^2+^ show that most strains coming from healthy people tend to form subclusters. When combining the cluster analysis of all elements, a similar outcome was observed, in which the majority of strains from healthy individuals are grouped together (7 out of 11).

To determine the existence of statistically significant differences between the group of strains coming from patients with IBD (Group 1) and the ones coming from healthy subjects (Group 2), inferential statistical analysis was carried out. To compare each variable between the two groups of interest, Mann–Whitney *U* test was applied. The null hypothesis for the test is that the distribution of the element is the same across the groups established. The null hypothesis was rejected for the concentrations of manganese and iron. The concentration of manganese was found to be significantly (*p* < 0.05) higher in the group of strains coming from healthy subjects: *U* = 2.619, *p* = 0.008. The concentration of iron is significantly higher in this group as well: *U* = 3.518, *p* = 1.898·10^–4^.

## 4. Discussion

The daily diet can contain a higher amount of H_2_S if certain food commodities are included in it (sulfur oxides, sulfate polysaccharides (mucin), chondroitin sulfate, carrageenan, etc.). The calculated sulfate in the western dies is over 16.6 mmol sulfate d^−1^ [49], while there is a higher chance (around 50%) that the feces of healthy individuals contain SRB (*Desulfovibrio*: up to 92%) [12,26]. It should be emphasized that higher H_2_S concentrations are toxic also for its producer. The growth of *Desulfovibrio* is stopped in the presence of H_2_S concentrations higher than 6 mmol L^−1^, though metabolic activity is not 100% inhibited. Lag phase and generation time at H_2_S concentrations of 5 mmol L^−1^ were two and eight times longer, respectively [8]. Clostridia can also produce H_2_S, though in smaller quantities [29]. Earlier studies with mice (specially conditioned animals for the experiment) showed 1.14 times higher H_2_S production among the samples of mice with UC in comparison with healthy animals [20,26].

Sulfate consumption, H_2_S production, lactate consumption and acetate accumulation play an important role in the environment, concerning the development of the intestinal bowel diseases [17,35,50,51]. *Desulfovibrio* spp. are related to the occurrence and development of the IBD since it can be very often found in the intestines and feces of people and animals with this ailment. SRB use sulfate as a terminal electron acceptor [21,50,51]. These facts can lead us to the conclusion that sulfate present in different food commodities (some bread, soy flour, dried fruits, brassicas and sausages, as well as some beers, ciders and wines) [49] could be an important factor in the development of bowel disease.

Principal component analysis grouped *Desulfovibrio* strains from individuals with colitis in one cluster according to the biomass accumulation and H_2_S production. The other cluster, with the same parameters, was formed from the strains found in healthy individuals. Between sulfate and lactate consumption was found a negative correlation (Pearson correlations, *p* < 0.01). Lower linear regression (*R*^2^) was found between biomass accumulation and H_2_S. Biomass accumulation and H_2_S production represent kinetic parameters and they play an important role in bowel inflammation, the same as in the development of ulcerative colitis. There is a probable synergy interaction between H_2_S and acetate produced by SRB since minor factors in the development of bowel disease are sulfate consumption and lactate oxidation [33]. A meta-analysis study showed higher SRB occurrence in patients with UC, showing that the production of H_2_S reaches toxic levels in IBD individuals, including UC [39].

The higher presence of electron acceptors and donor concentrations resulted in more intense growth of *D. piger* Vib-7. In this case, the accumulation of H_2_S and acetate was intense, too. These conditions lead to the occurrence of UC and consequently to bowel cancer. Adverse effects of H_2_S were noticed on the intestinal mucosa, epithelial cells, the growth of colonocytes [11,17,19,20,21,22,23,24]. H_2_S causes phagocytosis and the death of intestinal bacteria [11,19], the same as it induces hyperproliferation and epithelial cell metabolic abnormalities [19]. The colon inflammation is also connected with the presence of SRB and high levels of their metabolites [7,11,13]. The integrity of colonocytes is regulated by H_2_S concentrations [52,53]. In the samples collected from the people with UC, H_2_S production is higher [39]. The differences between samples from healthy and unhealthy individuals were noticed in our previous studies too [33]. SRB play an important role in bowel disease development, although the intestinal microbiota represents a complex system, where interactions are occurring between clostridia, methanogens, lactic acid bacteria, etc. [10,29,31,50]. On the other side, many intestinal microorganisms, including lactic acid bacteria and methanogens, can be inhibited by H_2_S [10,50].

Studies dealing with SRB presence in the intestines and correlation about the prevalence of intestinal bowel disease and H_2_S are very important since H_2_S-releasing agents can be seen as promising therapeutic agents [52]. H_2_S is also confirmed to be an important cardiovascular and nervous system signaling factor. By the conversion to thiosulfate cecal mucosa protects itself from H_2_S toxic effects [52,53,54].

## 5. Conclusions

SRB were present in the samples of healthy and unhealthy individuals. Mainly *Desulfovibrio vulgaris* was identified by sequence homology analysis of the 16S rRNA gene. It has to be emphasized that differences in cellular shape were not detected among SRB isolates, from both groups (healthy and unhealthy individuals). Isolated and purified strains formed one cluster *Desulfovibrio longreachensis* ACM 3958 and *Desulfovibrio vulgaris* subsp. Oxamicus DSM 1925. More distant clusters were grouped with other intestinal SRB species. Statistically significant differences were not found in biomass accumulation between samples obtained from healthy and unhealthy individuals. Biomass accumulated the most for Strain 7 (fivefold increase). Oppositely, statistically significant (*p* < 0.05) differences were observed among all measured parameters (sulfate, H_2_S and biomass), after 24 and 48 h of cultivation among healthy individuals and patients with IBD. The strains isolated from healthy people grouped mainly the lower clusters (clusters based on manganese). The main elements that distinguished samples, according to the detected trace element, were manganese and iron. These elements were found in higher concentrations among healthy individuals. Our study allows drawing a clearer picture about the microbial processes, which occur during IBD. The physiological differences among SRB strains of healthy and unhealthy individuals are emphasizing the importance of better understanding of microbiological processes involved in IBD. The study has revealed a clearer role of SRB by the combination of physiological, molecular biological and statistical analyses, and specific SRB ecotypes in the development of the intestinal bowel diseases.

## Figures and Tables

**Figure 1 jcm-09-01920-f001:**
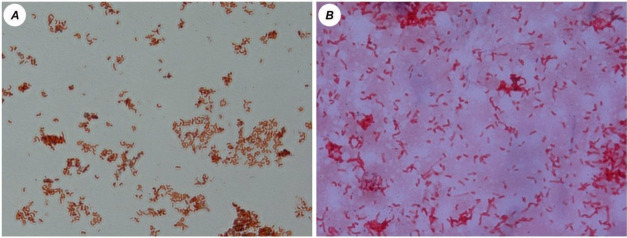
Morphological distributions of sulfate-reducing bacteria (SRB) mixed cultures from healthy people (**A**) and patients with inflammatory bowel diseases (IBDs, **B**; magnification × 1000).

**Figure 2 jcm-09-01920-f002:**
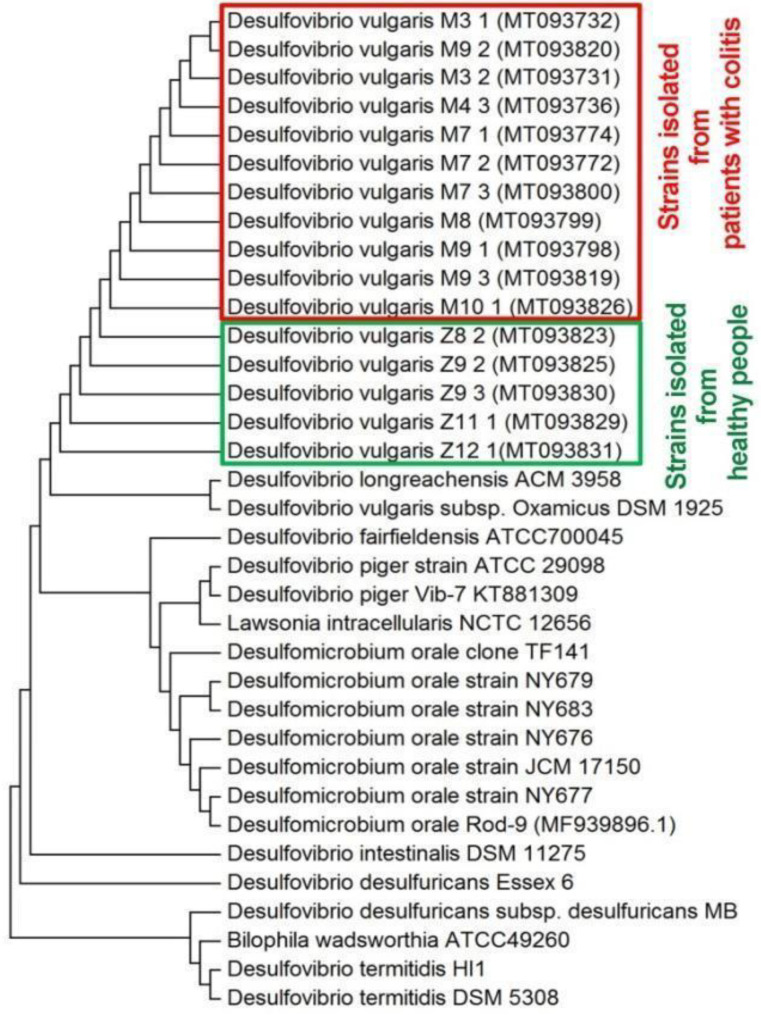
Phylogenetic tree based on sequences analysis of the 16S rRNA gene known and studied intestinal SRB strains (built by the neighbor-joining method).

**Figure 3 jcm-09-01920-f003:**
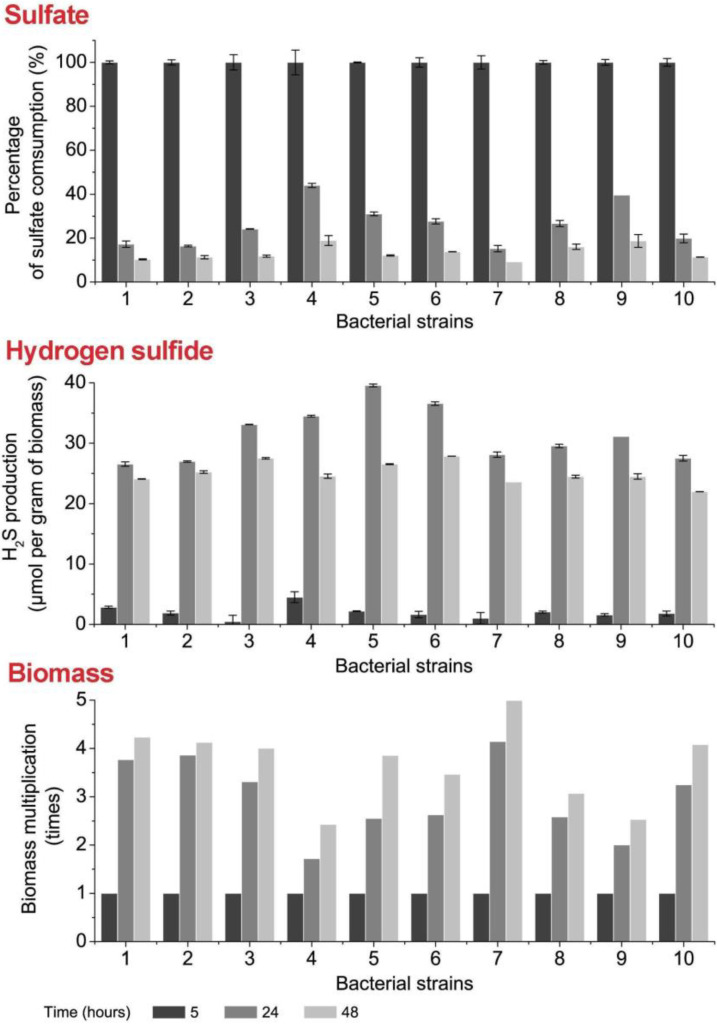
Sulfate consumption, H_2_S production and biomass accumulation by collected stains of SRB during 5, 24 and 48 h (**IBD**: **1**—M1.1, **2**—M7.3, **3**—M8, **4**—M9.2, **5**—M9.3; **healthy**: **6**—Z8.2, **7**—Z9.2, **8**—Z9.3, **9**—Z11.1, **10**—Z12.1).

**Figure 4 jcm-09-01920-f004:**
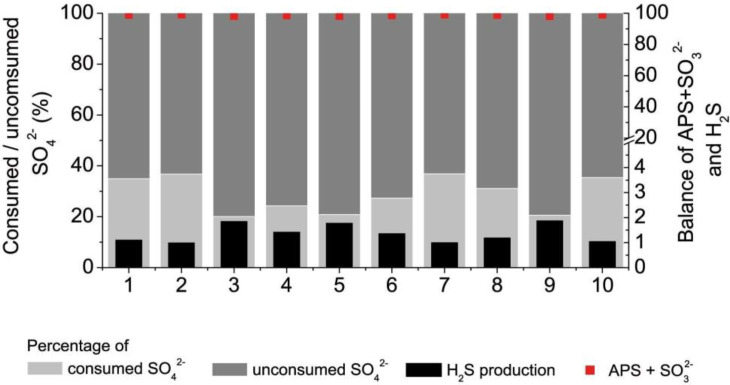
Balance of sulfate consumption, H_2_S production and APS/sulfite accumulation by collected stains of SRB within the first 24 h.

**Figure 5 jcm-09-01920-f005:**
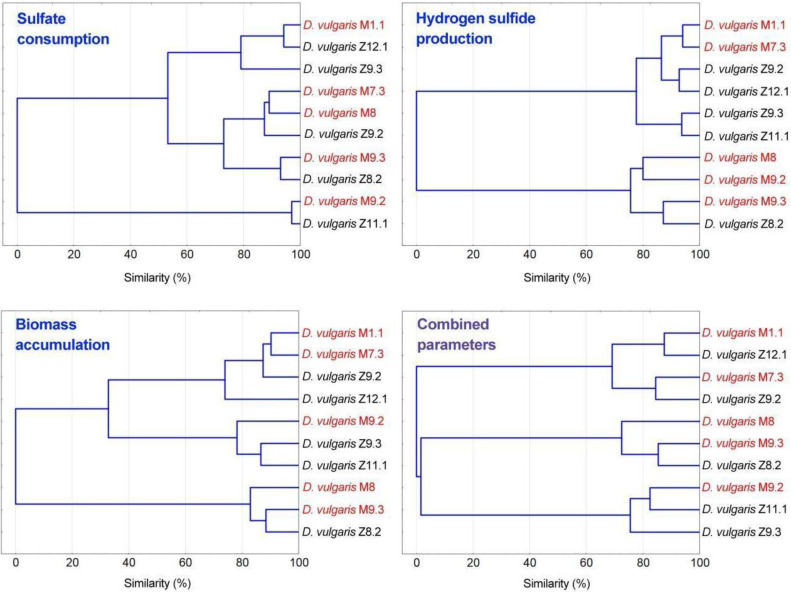
Cluster analysis of sulfate consumption, H_2_S production and biomass accumulation parameters.

**Figure 6 jcm-09-01920-f006:**
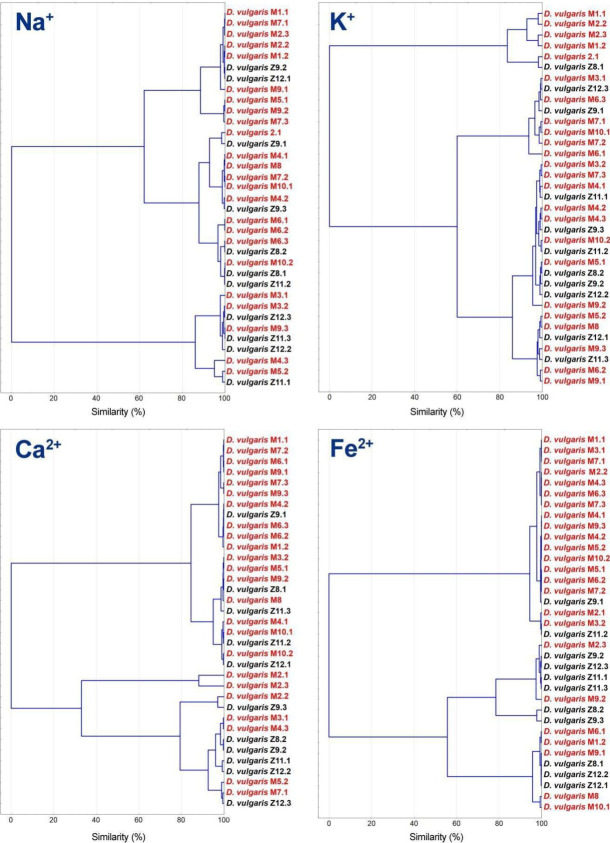
Cluster analysis based on concentrations data of physiological important elements (Na^+^, K^+^, Ca^2+^ and Fe^2+^).

**Figure 7 jcm-09-01920-f007:**
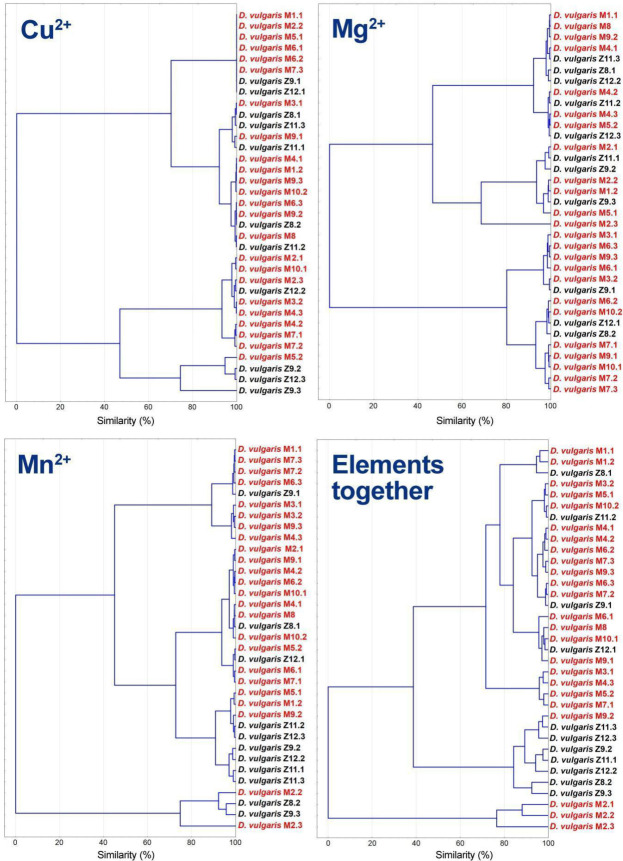
Cluster analysis based on concentrations data of physiological important elements (Cu^2+^, Mg^2+^, Mn^2+^ and all described elements in the previous and present figures).

**Table 1 jcm-09-01920-t001:** Patients and control subject’s characteristics.

Sample Name	Sex	Diagnosis	Age	Weight (kg)	Year of Diagnosis	State of the Disease *
M-01	Female	CD	21	57	2011	Short-term remission
M-02	Male	CD	34	104	2013	Short-term remission
M-03	Female	UC	66	78	2012	Short-term remission
M-04	Female	UC	41	53	2005	Disease flare
M-05	Male	UC	35	92	2018	Short-term remission
M-06	Female	UC	39	120	2013	Disease flare
M-07	Male	UC	44	61	2014	Long-term remission
M-08	Female	UC	20	70	2003	Short-term remission
M-09	Male	CD	24	73	2006	Long-term remission
M-10	Male	CD	34	70	2018	Disease flare
Z-8	Male	Control	46	108	–	Healthy
Z-9	Male	Control	53	72	–	Healthy
Z-11	Male	Control	42	95	–	Healthy
Z-12	Male	Control	26	78	–	Healthy

* The induction of remission is dated from for each sample as followed: 5/2019 (M-01), 10/2019 (M-02), 4/2019 (M-03), 10/2019 (M-05), 2/2015 (M-07), 7/2019 (M-08), 11/2017 (M-09).

**Table 2 jcm-09-01920-t002:** Prescribed and taken therapeutics assigned to IBD sample donors *.

Sample Name	Azathioprine	Mesalazine	Glucocorticoids	Biological Treatment	Omeprazole
M-01	Imuran 50 mg	–	–	–	–
M-02	–	Pentasa 500 mg	–	–	–
M-03	–	Pentasa 1 g	Cortiment 9 mg	–	Helic 10 mg
M-04	Imuran 50 mg	Pentasa 1 g	Prednison 5 mg	–	–
M-05	–	Pentasa 1 g	Cortiment 9 mg	–	–
M-06	–	Asacol 400 mg	Cortiment 9 mg	Entivio 300 mg	–
M-07	–	Pentasa 1 g	–	–	–
M-08	Imasup 50 mg	Salofalk 3 g	–	Remsima 400 mg	–
M-09	–	Pentasa 1 g	–	–	–
M-10	–	Pentasa 1 g	Budenofalk 3 mg	–	–

* The active compound in the table is assigned by commercial name, under which it was prescribed by the medical expert, mark (–) means that the therapeutics was not prescribed to the patient.

**Table 3 jcm-09-01920-t003:** Identified species of pure bacterial cultures by sequence analysis of the 16S rRNA gene.

Identified Bacteria and Accession in GenBank	Amplicon Length (bp)	% Identity	Reference Strains of *Desulfovibrio* Genus	Reference Accession in GenBank
*Desulfovibrio vulgaris* M6_1 MT027815	745	99.06	*D. vulgaris* DSM 644	NR_112657.1
			*D. vulgaris* RCH1	CP002297.1
			*D. vulgaris* DP4	CP000527.1
*Desulfovibrio vulgaris* M6_2 MT027923	740	98.65	*D. vulgaris* RCH1	CP002297.1
			*D. vulgaris* DSM 644	NR_112657.1
			*D. vulgaris* DP4	CP000527.1
*Desulfovibrio vulgaris* M6_3 MT027925	279	97.47	*D. vulgaris* PhM31	KC013878.1
			*D. vulgaris* PhM22	KC013874.1
			*D. vulgaris* Hildenborough	NR_074446.1
*Desulfovibrio vulgaris* M1_1 MT027899	288	93.57	*D. vulgaris* PhM31	KC013878.1 KC013874.1 CP002297.1
			*D. vulgaris* PhM22	
			*D. vulgaris* RCH1	
*Desulfovibrio vulgaris* M1_2 MT027900	575	99.30	*D. vulgaris* Hildenborough	NR_074446.1 CP002297.1 CP000527.1
			*D. vulgaris* RCH1	
			*D. vulgaris* DP4	
*Desulfovibrio vulgaris* M3_1 MT093732	391	99.22	*D. vulgaris* Hildenborough	NR_074446.1
			*D. vulgaris* DP4	CP000527.1
			*D. vulgaris* RCH1	CP002297.1
*Desulfovibrio vulgaris* M3_2 MT093731	429	99.30	*D. vulgaris* Hildenborough	NR_074446.1
			*D. vulgaris* RCH1	CP002297.1
			*D. vulgaris* DSM 644	NR_112657.1
*Desulfovibrio vulgaris* M4_3 MT093736	748	99.87	*D. vulgaris* RCH1	CP002297.1
			*D. vulgaris* DSM 644	NR_112657.1
			*D. vulgaris* DSM 644	NR_041855.1
*Desulfovibrio vulgaris* M5_1 MT093737	568	98.07	*D. vulgaris* PhM31	KC013878.1
			*D. vulgaris* PhM22	KC013874.1
			*D. vulgaris* Hildenborough	NR_074446.1
*Desulfovibrio vulgaris* M5_2 MT093788	745	99.46	*D. vulgaris* RCH1	CP002297.1
			*D. vulgaris* DSM 644	NR_112657.1
			*D. vulgaris* DP4	CP000527.1
*Desulfovibrio vulgaris* M7_1 MT093774	751	99.60	*D. vulgaris* RCH1	CP002297.1
			*D. vulgaris* DSM 644	NR_112657.1
			*D. vulgaris* DP4	CP000527.1
*Desulfovibrio vulgaris* M7_2 MT093772	762	99.47	*D. vulgaris* RCH1	CP002297.1
			*D. vulgaris* DSM 644	NR_112657.1
			*D. vulgaris* Hildenborough	AE017285.1
*Desulfovibrio vulgaris* M7_3 MT093800	766	99.61	*D. vulgaris* DP4	CP000527.1
		99.35	*D. vulgaris* RCH1	CP002297.1
			*D. vulgaris* DSM 644	NR_112657.1
*Desulfovibrio vulgaris* M8 MT093799	762	99.34	*D. vulgaris* RCH1	CP002297.1
			*D. vulgaris* DSM 644	NR_112657.1
			*D. vulgaris* DP4	CP000527.1
*Desulfovibrio vulgaris* M9_1 MT093798	768	99.60	*D. vulgaris* RCH1	CP002297.1
			*D. vulgaris* DSM 644	NR_112657.1
			*D. vulgaris* DP4	CP000527.1
*Desulfovibrio vulgaris* M9_2 MT093820	315	98.10	*D. vulgaris* PhM31	KC013878.1
			*D. vulgaris* PhM22	KC013874.1
			*D. vulgaris* Hildenborough	NR_074446.1
*Desulfovibrio vulgaris* M9_3 MT093819	764	99.73	*D. vulgaris* RCH1	CP002297.1
			*D. vulgaris* DSM 644	NR_112657.1
			*D. vulgaris* DP4	CP000527.1
*Desulfovibrio vulgaris* M10_1 MT093826	770	99.09	*D. vulgaris* RCH1	CP002297.1
			*D. vulgaris* DSM 644	NR_112657.1
			*D. vulgaris* DP4	CP000527.1
*Desulfovibrio vulgaris* Z8_2 MT093823	768	99.48	*D. vulgaris* RCH1	CP002297.1
			*D. vulgaris* DSM 644	NR_112657.1
			*D. vulgaris* DP4	CP000527.1
*Desulfovibrio vulgaris* Z9_2 MT093825	629	99.05	*D. vulgaris* RCH1	CP002297.1
			*D. vulgaris* DSM 644	NR_112657.1
			*D. vulgaris* DP4	CP000527.1
*Desulfovibrio vulgaris* Z9_3 MT093830	448	99.33	*D. vulgaris* RCH1	CP002297.1
			*D. vulgaris* DSM 644	NR_112657.1
			*D. vulgaris* DP4	CP000527.1
*Desulfovibrio vulgaris* Z11_1 MT093829	751	99.46	*D. vulgaris* DP4	CP000527.1
			*D. vulgaris. vulgaris*	DQ826728.1
		99.20	*D. vulgaris* RCH1	CP002297.1
*Desulfovibrio vulgaris* Z12_1 MT093831	463	99.13	*D. vulgaris* Hildenborough	NR_074446.1
			*D. vulgaris* RCH1	CP002297.1
			*D. vulgaris* DSM 644	NR_112657.1

**Table 4 jcm-09-01920-t004:** Statistical analysis of sulfate reduction parameters and biomass accumulation of intestinal SRB strains.

Parameters	Healthy People			Patients with IBD		
	5 h	24 h	48 h	5 h	24 h	48 h
Sulfate	23.23 ± 5.85 ^a^*	5.61 ± 0.99 ^b^	3.05 ± 0.32 ^d^	25.03 ± 5.34 ^a^	6.26 ± 1.72 ^bcd^	3.10 ± 0.33 ^cd^
H_2_S	1.62 ± 0.38 ^a^	30.58 ± 3.63 ^b^	24.48 ± 2.15 ^b^	2.38 ± 1.47 ^a^	32.14 ± 5.47 ^b^	25.58 ± 1.42 ^b^
Biomass	4.35 ± 1.07 ^a^	12.11 ± 1.35 ^bc^	15.04 ± 1.37 ^d^	4.02 ± 1.17 ^a^	11.55 ± 1.96 ^c^	14.32 ± 0.82 ^bd^

* different letters (a, b, c, d) indicated statistically significant (*p* < 0.05) difference.

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
