# Peer review of "Evaluation of Physiological Parameters of Intestinal Sulfate-Reducing Bacteria Isolated from Patients Suffering from IBD and Healthy People"

_jcm, 2020, doi:10.3390/jcm9061920_

Round 1

Reviewer 1 Report

In Table 1 the state of disease needs to be clarified. No patient with inflammatory bowel disease is in "permanent remission" which implies cure. Long-term remission is a better term. It should also be clarified as to what medication these patients were on at the time of the measurements in a note to the Table. "Temporary remission" would probably be better called short-term remission. The duration of both forms of remission should be defined as a note to the table.

Author Response

In Table 1 the state of the disease needs to be clarified. No patient with inflammatory bowel disease is in "permanent remission" which implies a cure. Long-term remission is a better term. It should also be clarified as to what medication these patients were on at the time of the measurements in a note to the Table. "Temporary remission" would probably be better called short-term remission. The duration of both forms of remission should be defined as a note to the table:

It was corrected and additional information was added. The terminology of the state of the disease was in fact confusing. It was based on the translation of currently used methodology in the Czech republic FNO and it was, as you pointed out, prone to misinterpretation. We corrected the table according to your suggestions, which better reflect the disease course and recent terminology. The remission induction dates are listed below in Table 2, which in our opinion better reflects the term that definition which might generalize it. Since the patients were undergoing various therapeutic plans, we enclosed Table 2 in which the main classes of therapeutics, as well as their brand names and concentrations, are enlisted with the proper description of table meaning. We hope that this clarifies the matter so that our article is clear of confusion and understandable.

Reviewer 2 Report

In this study the authors identified sulfate-reducing bacteria in patients suffering from IBD and in healthy individuals and investigated different parameters. The dominant strain found in both groups was Desulfovibrio vulgaris. Although no significant differences concerning the compostion of the intestinal microbiota was found the authors observed differences in the manganese and iron contents which were higher in the healthy control group compared to IBD patients.

Methods are described in detail.

Comments:

1) Only 4 persons are included for the healthy individuals. There should also be ten as in the group of patients suffering from IBD.

2) The authors investigated ten bacterial strains out of 35 isolated strains to investigate different Parameters (Fig 3). What kind of criterions were used to choose these ten strains? A correlation between the 35 isolated strains and the ten analyzed strains should be made.

3) What is missing within the whole results is an annotation of the different strains. It is not clear which strain belongs to healthy individuals and which to IBD patients. The aim of the study is to compare healthy people and IBD patients concerning their microbiota. In the present way there is only made a correlation between the different strains but not between the patients suffering from IBD and healthy people. All figures should be rearranged to point out the main declaration.

Author Response

In this study the authors identified sulfate-reducing bacteria in patients suffering from IBD and in healthy individuals and investigated different parameters. The dominant strain found in both groups was Desulfovibrio vulgaris. Although no significant differences concerning the composition of the intestinal microbiota was found the authors observed differences in the manganese and iron contents which were higher in the healthy control group compared to IBD patients.

Methods are described in detail.

Comments:

1) Only 4 persons are included for healthy individuals. There should also be ten as in the group of patients suffering from IBD.

We agree with the reviewer that more healthy individuals would give an even more clear picture of the experiment. Though, the experiment was designed in the way that emphasis was put on differences among patients with IBD. The samples from healthy individuals were taken due to the comparison with individuals with developed bowel disease. We hope that you accept our study design.

2) The authors investigated ten bacterial strains out of 35 isolated strains to investigate different Parameters (Fig 3). What kind of criteria were used to choose these ten strains? A correlation between the 35 isolated strains and the ten analyzed strains should be made.

At the beginning of the experiment there were 35 strains of SRB isolated, both from healthy and unhealthy individuals. According to their physiological properties we chose 10 SRB strains, which growth, sulfate reduction and hydrogen sulfide production were measured.  The reviewer is right that the evaluation of 35 SRB strains would give more clear findings, but we have to admit that the evaluation of all 35 strains would have been more than challenging, as the provided personal and technical capacity at the Department was unfortunately limiting at the time of the study. We are also very unfortunate with this minor gap. However, the cluster analysis of all 35 strains was done based on concentrations data of physiological elements. These findings showed us the vicinity of different strains, isolated from healthy and unhealthy individuals. We are planning to continue the evaluation in further studies.

3) What is missing within the whole results is an annotation of the different strains. It is not clear which strain belongs to healthy individuals and which to IBD patients. The aim of the study is to compare healthy people and IBD patients concerning their microbiota. In the present way there is only made a correlation between the different strains but not between the patients suffering from IBD and healthy people. All figures should be rearranged to point out the main declaration.

It was corrected. Now we hope you find it possible to distinguish between the evaluated strains in the experiment.

Round 2

Reviewer 2 Report

The comments were completely answered and the manuscript is now accaptable in it`s present form.

Author Response

Reply to reviewer 2 comments

The comments were completely answered and the manuscript is now accaptable in it`s present form.

Reply: Thank you very much for performing the review!